# Bacterial uropathogens and burden of antimicrobial resistance pattern in urine specimens referred to Ethiopian Public Health Institute

Tesfa Addis[1], Yonas Mekonnen[1], Zeleke Ayenew[1], Surafel Fentaw[1], Habtamu Biazin[2]*

1 Department of Clinical bacteriology and Mycology, Ethiopian Public Health Institute, Addis Ababa, Ethiopia, 2 Department of Microbiology, Immunology and Parasitology, College of Health Sciences, Addis Ababa University, Addis Ababa, Ethiopia

☯ These authors contributed equally to this work.
* habtamu.biazin@aau.edu.et

## Abstract

### Background

Urinary tract infections (UTIs) are the leading causes of morbidity in the general population, and is the second most common infectious disease after respiratory infections. Appropriate antibiotic therapy is essential to achieving good therapeutic results. Therefore, the purpose of this study was to investigate the profile of pathogens cultured from urinary tract infections and to determine their resistance profiles to commonly prescribed antibiotics.

### Method

A cross-sectional study was carried out at the National Referral Laboratory of the Ethiopian Institute of Public Health from January 2017 to December 2018. All positive cultures were characterized by colony morphology, Gram stain, and standard biochemical tests. The antimicrobial susceptibility test of the isolate was performed using the Kirby- Bauer disk diffusion test on Muller-Hinton agar. In addition, bacterial identification, antimicrobial susceptibility testing and phenotypic detection of MDR were performed with VITEK 2 Compact according to the manufacturer's instructions.

### Result

Out of 1012 cultured urine specimens, 325 (32.1%) was showed significant bacteriuria. The overall prevalence of UTIs was 325(32.1%) and the highest prevalence rate was obtained from 21–30 years age group 73(22.5%). Among UTIs patients, 583(57.6%) were females and 429(42.4%) were males. The UTIs of 179 (55%) women is relatively higher than that of men 146 (45%). Among 325 isolates, Gram-negative bacteria (GNB) appeared more frequently 252 (51.7%) than Gram-positive bacteria 63 (19.4%).

In GNB, *E. coli* 168(66.7%), *Klebsiella species* 32(12.7%), and *Enterobacter species* 13 (5.2%) were dominated isolates whereas in GPB accounted for coagulase-negative

**Funding:** The author(s) received no specific funding for this work.

**Competing interests:** The authors have declared that no competing interests exist.

**Abbreviations:** CoNS, Coagulase Coagulase-negative staphylococcus; GNB, Gram-negative bacteria; GPB, Gram-positive bacteria; MDR, Multidrug-resistant; MDRO, Multidrug resistant organism; PDR, Pandrug resistant; UTI, Urinary tract infection; XDR, Extensive drug-resistant.

staphylococcus (CoNS) 33(52.4%), *Enterococcus species* 16(25.4%), and *Staphylococcus aureus* 10(15.9%). Major of the isolates showed high levels of antibiotic resistance to commonly prescribed antimicrobials. Imipenem, Amikacin, and Nitrofurantoin were the most sensitive antibiotics for Gram-negative isolates while Nitrofurantoin, clindamycin, and Gentamycin were effective against gram-positive uropathogens. Overall, 156/256(60.9%), 56/256(22.4%), 10/256(4%) of gram-negative isolates were MDR, XDR, and PDR respectively while among the GPB isolates, 34/63(53.1%), 10/63(15.8%), and 1/63(1.6%) were MDR, XDR, and PDR isolates respectively. Among the tested bacterial strains, 190/319 (59.5%) were MDR, 66/319 (20.7%) strains were XDR, and 11/319 (3.45%) were PDR isolated.

## Conclusion

The prevalence of urinary tract infection was high, and Gram-negative organisms were the most common causes of UTIs in this study. It was found that the resistance to commonly used antibiotics is very high. Early detection and close monitoring of MDR, XDR, or even PDR bacterial strains must be started by all clinical microbiology laboratories to reduce the menace of antimicrobial resistance that is now a global problem.

## Introduction

Urinary tract infections (UTIs) are infections caused by the presence, growth and spread of microorganisms in the urinary tract. It is usually caused by bacteria from the digestive tract climbing into the urethra opening and starting to multiply to cause an infection. It is one of the most common bacterial infections seen in clinical practice in developing countries [1]. Urinary tract infections are among the most common infections in humans, exceeded in frequency among ambulatory patients only by respiratory and gastrointestinal infections. It is believed to exist when pathogenic microorganisms are detected in the urine, urethra, bladder, kidneys, or prostate with or without the presence of specific symptoms [2].

The clinical presentations of UTIs depend on the part of the urinary tract affected, the causative pathogens, the severity of the infection, and the patient's ability to mount an immune response to it [3]. An infection in the bladder is called cystitis or a bladder infection. Infection of one or both kidneys is called pyelonephritis or a kidney infection. The ureters are tube-like structures that take urine from each kidney to the bladder are only rarely the site of infection. The urethra, that empties urine from the bladder to the outside and its infection is called urethritis [1,4].

Intestinal gram-negative bacteria are a serious urinary tract infection problem in many parts of the world. Symptomatic urinary tract infections (UTIs) have been estimated to occur with approximately seven million emergency department visits and 100,000 hospital admissions each year [1]. Urinary tract infections have become the most common nosocomial infection, accounting for 35% of nosocomial infections and the second leading cause of sepsis in hospitalized patients [5,6].

The causative agent of urinary tract infections may be of community or hospital origin. Community-based infections are caused by *Escherichia coli*, *Klebsiella pneumoniae*, *Proteus mirabilis*, *Staphylococcus saprophyticus*, or *Enterococcus faecalis*, while nosocomial infections are *Escherichia coli*, *Pseudomonas aeruginosa*, *Proteus*, *Enterobacter*, *Serratia*, or *Enterococcus*.

[5]. Everyone is susceptible to UTIs; however, some traits increase exposure to uropathogens and others increase susceptibility to the development of symptoms after colonization.

Factors that increase exposure to uropathogens include the presence of urinary catheters and vaginal intercourse; markers of host susceptibility to the disease include age; Kind; and the presence of underlying conditions affecting the urinary tract, such as pregnancy, diabetes, or an enlarged prostate. Bacterial characteristics increase the ability of the agent to be transmitted, cause infection, and cause disease [3]. Recurrent infections are common and can lead to irreversible kidney damage, leading to renal hypertension and renal failure in severe cases [3].

UTI is the most common cause of nosocomial infection among hospitalized patients [6]. It is generally accepted that UTI can only be diagnosed based on a microscope and microbial culture. Urine culture is the gold standard for assessing infection, but in a majority of cases, therapy is initiated before having culture results. This can be rationalized in the sense that urine culture and susceptibility testing are costlier that than the costs of antibiotics in most countries [3].

In most cases, it is necessary to start treatment before the final microbiological results are available. The test strip/slide method is used in many centres only as a screening method, but cultures are required for the final diagnosis [7,8]. Region-specific surveillance studies that gather knowledge about UTIs and their resistance patterns can help clinicians choose the most appropriate empiric treatment. Antibacterial drugs are helpful in the management of bacterial infections. Although some bacteria have inherent resistance to even recently formulated antibacterial agents, the emergence of acquired antimicrobial resistance has been observed in most pathogens [9–11].

Antibiotic resistance has become a significant public health problem with grave consequences for the treatment of infections. Over the past decade, there has been an increasing prevalence of carriage and infection with multidrug-resistant organisms (MDROs). This ultimately affects both social and economic development. Multidrug resistance (MDR) has been increased all over the world that is considered a public health threat. Several recent investigations reported the emergence of multidrug-resistant bacterial pathogens from different origins including humans [12], poultry [13], cattle [14,15], and fish [16,17] that increase the need for routine application of antimicrobial susceptibility testing to detect the antibiotic of choice as well as the screening of the emerging MDR strains [12–14,18]. Infections caused by MDR organisms have increased mortality compared to those caused by non-MDR bacteria. The problem is attributed to the misuse of antimicrobials, which create selective pressures that facilitate the emergence of resistant strains [10].

To prevent the problem of drug resistance, the World Health Organization (WHO) has put in place several interventions. These include the creation of a national task force, the development of indicators to monitor and assess the impact of antimicrobial resistance; and designing microbiological baselines capable of effectively coordinating the surveillance of antibiotic resistance among common pathogens [19,20]. Although these interventions appear to be well implemented in the developed countries, lack of resources has limited their implementation in many developing countries where treatment options also tend to be relatively limited.

Thus, although this is a global problem, antimicrobial resistance tends to be more important in developing countries than in developed countries [21]. Studies in the eastern part of Nepal, India, and Bangladesh have reported an increased resistance of urinary pathogens to commonly used antibiotics [22–24]. The magnitude of UTI-causing organisms and their sensitivity to antibiotics varies from region to region, so antibiotic resistance and sensitivity patterns need to be tested frequently. The causative agents and degree of resistance to the most commonly prescribed drugs used in the treatment of UTIs may also have changed over time. Therefore, continuous and periodic monitoring of the local prevalence of uropathogens and

their susceptibility profile will be of public health importance to promote the appropriate use of existing antibiotics [3]. Therefore, this study was performed to determine the frequency and antibiotic susceptibility pattern of urinary tract pathogens received in the Ethiopian public health institute.

## Methods

### Study design and study period

A cross-sectional study was conducted from January 2017 to December 2018 in Addis Ababa, Ethiopia.

### Study setting

The study was conducted at Ethiopian Public Health Institute, which is a national reference laboratory in Addis Ababa, Ethiopia.

### Source population

All urine samples were referred to the Department of Clinical Microbiology of the Ethiopian Institute of Public Health for culture and sensitivity testing during the study period (January 2017 to December 2018).

### Study population

All the urine samples obtained between January 2017 and December 2018 for urine culture and susceptibility testing from patients of all ages were included in this study.

**Inclusion criteria.**   All data with complete information such as age, sex, number of patients, pathogens isolated, antibiotic resistances of isolates, were included in the study.

**Exclusion criteria.**   Data with incomplete information was excluded from the study.

### Sample size determination and sampling

A total of 1012 urine samples were collected between January 2017 and December 2018 at EPHI national microbiology referral Laboratory.

### Ethical review

The ethical review was first obtained from the ethics and review committee of the Menelik II Health Science College and Ethiopian public health. Formal written letters have been distributed to the Department. Before collecting data, study participants were informed of the study and obtained their consent, and confidentiality was maintained by omitting their names and personal identifiers throughout the study.

The research objectives and procedures were explained in writing (if they can read and write) or verbally (if they are illiterate), and the children and their parents or caregivers were asked if they would like the children to participate in the research. Parents or caregivers confirmed their children's willingness to participate in the study by signing an informed consent form. Confidentiality was strictly maintained during data processing and report writing.

### Isolation, identification and characterization of bacteria causing UTIs

**Isolation of pathogens.**   The pathogens have been isolated from urine sediments. The urine sample was shaken well to resuspend the organisms and 10 ml was decanted into a centrifuge tube. The tube was kept closed to avoid contamination [25]. The sample was

centrifuged at 2000 rpm for 10 minutes. The entire sample was decanted, but 0.5 ml of sediment from the tube was suspended with a sterile metal loop. A loop of the sediment was inoculated into a tube with a medium [26].

## Identification of pathogens

**Cultural observation.** Color, size, and colony morphology are observed from the incubated plates.

**Microscopic examination of urine specimen.** Slides were prepared from each different colony observed on the plates and Gram staining was performed. Results such as arrangement, gram reaction (gram-positive or gram-negative), and shape of bacteria are seen from the investigations [25,27]

## Microbiological analysis of urine specimen

Uropathogens were identified by inoculating /streaking of urine samples on various selective and differential media such as CLED agar, blood agar, chocolate agar, MacConkey agar based on their colour morphology after an incubation time of 18–24 hours at 37˚C [26].

## Biochemical examination

The selected colonies were subjected to culture-based biochemical examinations, microscopic and microbiological examinations (Oxoid Ltd, Basingstoke, UK) (carbohydrate utilization test, triple sugar iron agar test or Kligler iron agar), oxidase test, catalase test, nitrate reduction test, indole production test, methyl red test, Voges-Proskauer test, citrate recovery test, urease test) for pathogen detection [27,28].

## Antimicrobial susceptibility testing

Antibiotic susceptibility testing of the isolates was performed by the Kirby–Bauer disk diffusion test on Mueller–Hinton agar for the following antimicrobial agents (Oxoid, Basingstoke, and Hampshire, UK). Gram-positive isolates were tested for Ciprofloxacin (CIP 5 μg), clindamycin (CLN 2 μg), Nitrofurantoin (NIT 300μg), Trimethoprim-sulphametazol (SXT 23μg), Norfloxacin (Nor 10μg), Tetracycline (TE 30 μg), Oxacillin (OX 1μg), Erythromycin (ERY 15 μg), Penicillin (PEN 10 μg), Vancomycin (Van), and Gentamycin (Gen 10μg) [29].

Gram-negative isolates were tested for Amikacin (AMK 30μg), Gentamicin (GEN 10μg), Ceftazidime (CAZ 30μg), Nitrofurantoin (NIT 300μg), Cefazolin (KZ 5μg), Ceftriaxone (CRO 30μg), Cefuroxime (CXM 30μg), Ciprofloxacin (CIP 5μg), Piperacillin (PIP 100μg), piperacillin + Tazobactam (TZP 100μg), Trimethoprim-Sulfamethoxazole (SXT 23μg), Tetracycline (TE 30μg), Imipenem, (IMP 10μg), and Amoxicillin-clavunate (AUG 20/10μg) [29].

**Data analysis.** The collected data was sorted in the prepared spreadsheet and entered into EpiData v4.6. Then, the data was entered into SPSSv.25 software, which was used for data analysis. The laboratory results are summarized and presented by age groups, sex and isolation types. The prevalence rate of the isolates, frequency, susceptibility patterns and other descriptive statistics were computed. Percentages and ratios were presented in the tables. P-value ($p < 0.05$) was considered statistically significant.

**Quality control.** Data were collected from the EPHI Microbiology Laboratory for processing, analyzing and determination of prevalence uropathogens, and drug susceptibility profiles of uropathogens. The completeness, correctness, clarity and consistency of the data collected were checked before being included in the data entry form.

**Table 1. Age and sex distribution of patients referred to EPHI.**

| Variables | | Status of UTI | | Total | CI at 95% | P-value |
|---|---|---|---|---|---|---|
| | | Bacterial growth | No growth | | | |
| Age group | 1–10 | 12 | 40 | 52 | Reference (1) | .000 |
| | 11–20 | 17 | 37 | 54 | 0.90(0.16–5.1) | .905 |
| | 21–30 | 89 | 194 | 283 | 1.41(0.30–7.6) | .711 |
| | 31–40 | 38 | 157 | 195 | 1.42(.3–7.0) | .699 |
| | 41–50 | 48 | 84 | 132 | .73(.14–3.74) | .702 |
| | 51–60 | 46 | 94 | 140 | 1.71(.33–8.83) | .519 |
| | 61–70 | 39 | 59 | 98 | 1.47(.29–7.6) | .646 |
| | 71–80 | 30 | 20 | 50 | 1.98(.38–10.3) | .416 |
| | 81–90 | 2 | 6 | 8 | 4.5(.8–24.6) | .082 |
| | Total | 325 | 687 | 1012 | | 0.178 |
| Sex | Male | 146 | 283 | 429 | Reference(1) | 0.000 |
| | Female | 179 | 404 | 583 | 0.98(0.74–1.29) | 0.862 |
| | Total | 325 | 687 | 1012 | | 0.223 |

## Results

### Demographic characteristics

Of the 1028 patient data recorded, 1012 laboratory test results were obtained and analyzed. Due to their incomplete information, 16(1.6%) of the participants' data were excluded from the study. The majority of the study participants were females, which accounts for 583 (57.6%). The mean age of the study participants was 40.17 with a standard deviation of ±18.57.

The highest prevalence rate was recorded from 21–30 years age group 89/194(45.9%). However, this difference is not statistically significant (p = 0.178). Females 176(17.4%) UTI is relatively higher than in males 145(14.3%) with a non-significantly associated (p = 0.223) (Table 1).

### The magnitude of urinary tract infection and isolated uropathogens

The overall prevalence of urinary tract infection was 325(32.1%) and the highest prevalence rate was obtained in the 21 to 30 years-old age group of 73(22.5%). Females 179(55%) UTI is relatively higher than in males 146(45%). Among the 325 isolates obtained, the frequency of occurrence of gram-negative bacteria was higher than that of gram-positive bacteria. The most frequently isolated bacteria were *Escherichia coli* 168(51.7%), followed by CoNS 33(10.2%) and *Klebsiella species* 32(9.8%). Other urinary tract pathogens cause a significant UTI burden includes *Enterobacter species* 13(4%), *Acinetobacter species* 12(3.7%), *Citrobacter species* 12 (3.7%); *Pseudomonas aeruginosa* 8 (3.5%); *Proteus species* 4(1.2%); *Morganella morgani* 2 (0.6%) and *Edwardsella species* 1(0.4%).

GNB accounted for 63 (19.3%) of the isolates. The most important Gram-positive isolates were CoNS 33(10.1%), Enterococcus *species* 16(4.9%), *Staphylococcus aureus* 10 (3.1%), and *streptococcus agalactiae* 4(1.2%). It was also found that the isolation rates of *E. coli*, *Klebsiella species*, and CoNS were higher in isolates that were entirely from females. Candida species account for 10 (3.1%) of the total isolates (Table 2).

### Antimicrobial susceptible profiling

Among the tested gram-positive isolates, 70%, 81%, 69.7%, and 50% were relatively sensitive to Nitrofurantoin, clindamycin, Gentamycin, and vancomycin, respectively. As tabulated in

**Table 2. Distribution of uropathogenic microorganisms among patients with urinary tract infection.**

| Isolated uropathogens | Number of isolates n (%) |
|---|---|
| No growth | 687(67.9) |
| Significant growth ($10^5$ CFU/ml) | 325(32.1) |
| **Gram-negative** | **252(77.5)** |
| *Escherichia coli* | 168(51.7) |
| *Klebsiella species* | *32(9.8)* |
| *Enterobacter species* | 13 (4.0) |
| *Acinetobacter species* | 12(3.6) |
| *Citrobacter species* | 12(3.6) |
| *Pseudomonas aeruginosa* | 8(2.4) |
| *Proteus species* | 4(1.2) |
| *Morganella morgana* | 2(0.6) |
| *Edwardsella species* | 1(0.3) |
| **Gram-positive** | **63(19.4)** |
| *Staphylococcus aureus* | 10(3.1) |
| **Coagulase negative staphylococcus** | **33(10.2)** |
| *Enterococcus species* | 16(4.9) |
| *streptococcus agalactiae* | 4(1.2) |
| **Yeast** | **10(3.1)** |
| Candida species | **10(3.1)** |
| **Total** | **325(100)** |

Table three, erythromycin 15(62.5%), ciprofloxacin 34(74%), penicillin 17(66.7%), Tetracycline 21(65.6%), Oxacillin 12(48%), and Norfloxacin 21(75%) resistance rate was observed for the tested isolates. However, low rates of resistance were shown to Clindamycin (18.8%), Nitrofurantoin (29%), and gentamycin (30%). Among Gram-positive isolates tested, the drug resistance rate of S.*aureus* was a high 41(68.3%), followed by *Enterococcus species* 31(63.3%). The total resistance rate of Gram-positive bacterial isolates was 51% (Table 3).

The gram-negative isolates were mainly sensitive to amikacin 79(85%), Imipenem 84 (86.6%), Nitrofurantoin 89 (82%), and Gentamicin 141 (63.1%). Gram-negative bacteria were highly resistance to piperacillin 50(82%), Trimethoprim-sulfamethoxazole 130(79%), augmentin (amoxicillin-clavulanate) 95(63.8%), ceftriaxone 87(58.4%), cefuroxime 106 (69.7%), 44 (51.2%), ciprofloxacin 144(62.8%) for the tested isolates but demonstrated low-level of resistance to Imipenem (13.4%), Amikacin (15%), and Nitrofurantoin (18.3%). The overall resistance rate was 54.4%, which means that more than 50% of isolates were resistant to antibiotics tested (Table 4).

## MDR, XDR, and PDR resistance isolates and antibiogram

The antibiotic susceptibility profile of 1012 bacterial isolates was studied. Among the tested bacterial strains, 190/319(59.5%) were MDR, 66/319 (20.7%) strains were XDR, and 11/319 (3.45%) were PDR isolated.

Among tested GNB isolates, 156/256(60.9%) were MDR, 56/256(21.9%) were XDR, and 10/256(3.9%) were PDR respectively. Thirty-two (12.5%) of the isolates were sensitive to all classes of antibiotics. However, the results of multi-drug resistance comparison within species showed that the Species of *Enterobacter* 10/13(76.9%) were multi-drug resistant, 11/12 (91.7%) of the isolates of the Acinetobacter species were multi-drug resistant, and 99/168 (58.9%) of *Escherichia coli* isolates were multi-drug resistant. Likewise, 30/168(18%) of *Escherichia coli*, 9/22

**Table 3. Resistance profiling of gram-positive bacteria identified from UTI patients.**

| Antimicrobial tested | | Number of Isolates n = 63, | | | | | | | | |
|---|---|---|---|---|---|---|---|---|---|---|
| Classes | Antibiotics | *S.aureus (n = 10)* | | CoNS (n = 33) | | *Enterococcus spp* (n = 16) | | *S.agalactiae* (n = 4) | | *Total* |
| | | Total | R (%) | Total | R (%) | Total | R (%) | Total | R (%) | R (%) |
| Nitrofurantoin | FM | 6 | 0 | 17 | 4(23.5) | 5 | 3(60) | 3 | 0 | 7(29) |
| Aminoglycosides | GEM | 9 | 4(44.4) | 21 | 5(24) | - | - | 3 | 1(33.3) | 10(30.3) |
| Glycopeptide | VAN | - | - | - | - | 2 | 1(50) | 4 | 2(50) | 3(50) |
| Penicillin | OX | 4 | 4(100) | 18 | 6(31.6) | - | - | 3 | 2(66.7) | 12(48) |
| | PEN | 3 | 3(100) | 10 | 9(90) | 4 | 2(50) | 4 | 3(75) | 17(66.7) |
| Quinolones | CIP | 6 | 5(83.3) | 23 | 17(74) | 14 | 10(71.4) | 3 | 2(66.7) | 34(74) |
| | NOR | 7 | 7(100) | 9 | 7(77.8) | 8 | 4(50) | 4 | 3(75) | 21(75) |
| Lincosamides | CLN | 5 | 2(40) | 11 | 1(9.1) | - | - | - | - | 3(18.8) |
| Macrolides | ERY | 6 | 4(66.7) | 11 | 7(63.6) | 4 | 2(50) | 3 | 2(66.7) | 15(62.5) |
| Sulfonamides | SXT | 10 | 8(80) | 24 | 17(71) | - | - | - | - | 25(73.5) |
| Tetracycline | TE | 4 | 4(100) | 12 | 5(41.7) | 12 | 9(75) | 4 | 3(75) | 21(65.6) |
| Overall AMR for a bacterium | | 60 | 41(68.3) | 156 | 79(50.6) | 49 | 31(63.3) | 31 | 18(58) | (51%) |
| Overall AMR for isolates | | Total tested isolates 337 Resistance isolates 169 (51%) | | | | | | | | |

Key: Ciprofloxacin (CIP), Clindamycin (CLN), Nitrofurantoin (FM), Trimethoprim-sulphametazol (SXT), Norfloxacin (Nor), Tetracycline (TE), Oxacillin (OX), Erythromycin (ERY), Penicillin (PEN), Vancomycin (VAN) and Gentamycin (GEN), Total = Total number of tested isolates, R = resistance isolates in percentage, AMR = Antimicrobial resistance.

(41%) of *Klebsiella pneumoniae*, 3/8(37.5%) of *Klebsiella oxytoca*, 7/12(58.3%) of *Acinetobacter spp*, *and* 2/12(17%) of *Citrobacter spp were* extensive drug-resistant isolates (Table 5). *Overall*, 56/256(22.4%) of gram-negative isolates were XDR. In this study, 10/256(3.9%) of gram-negative isolates were Pandrug resistant. The majority of the PDR was attributed to *Escherichia coli (4), Acinetobacter spp (3), Enterobacter spp (1), and Citrobacter spp (1).*

Among 156 GNB-MDR strains isolated, the commonest MDR strains were detected from *Acinetobacter spp* 11/12(91.7%), followed by *Klebsiella pneumoniae* 17/22 (77%), *Enterobacter spp10/13(77), Klebsiella oxytoca 6/8(75%)* and *Citrobacter spp 9/12(75%).*

Among the GPB isolates, 34/63(53.1%) were multidrug resistance, 10/63(15.8%) were XDR and 1/63(1.6%) were PDR resistance isolates. Nine isolates (14.2%) of gram-positive bacteria were sensitive to all classes of antibiotics tested. However, within the tested isolates 8/10 (80%) of *staphylococcus aureus* were found to be multidrug-resistant, 6/10(60%) were XDR isolates. In this study, 17/29(58.6%) of *coagulase-negative staphylococcus* (CoNS) isolates were MDR, and 1/29(3.5%) were PDR. However, there was no XDR strain detected in the CoNS isolate. Among 34 MDR GPB isolates, 17 were CoNS; eight were *Staphylococcus aureus* and seven were *Enterococcus species* isolates. Similarly, out of 10 GNB-XDR strains isolated, the commonest XDR strains were detected from *Staphylococcus aureus* 6/10(60), followed by *Enterococcus spp* 3(18.8), and *S.saprophyticus* ¼(25%) as tabulated in Table 5.

## Discussion

Due to the high incidence of infection in the community and hospital environment, urinary tract infections have placed a huge burden on the health system [30]. Effective treatment of patients with bacterial urinary tract infections is usually based on the identification of the pathogen and the selection of effective antibiotics through continuous monitoring of the antimicrobial susceptibility pattern of urinary tract pathogens in specific areas [8]. The results of the

**Table 4. Resistance profiles of gram-negative isolate uropathogens.**

| Classes | Antibiotics | *E. coli* (N = 168) | | *Klebsiella* species (n = 32) | | *Enterobacter* species(n = 13) | | *Acinetobacter* species(n = 12) | | *Citrobacter* species(n = 12) | | *Pseudomonas* species(n = 8) | | Total R% |
|---|---|---|---|---|---|---|---|---|---|---|---|---|---|---|
| | | T | R | T | R | T | R | T | R | T | R | T | R | |
| Aminoglycosides | AMK | 51 | 0 | 13 | 2(15.4) | 10 | 6(60) | 8 | 3(37.5) | 9 | 1(11.1) | 2 | 2(100) | 14(15) |
| | GEN | 156 | 48(31) | 31 | 16(51.6) | 9 | 8(88.9) | 12 | 8(66.7) | 7 | 2(28.6) | 7 | 0 | 82(36.9) |
| Beta-Lactams | PIP | 32 | 28(87.5) | 10 | 8(80) | - | - | 6 | 6(100) | 10 | 8(80) | 3 | 0 | 50(82) |
| | CAZ | 52 | 25(48) | 7 | 4(57.1) | 6 | 3(50) | 8 | 7(88) | 6 | 4(67) | 7 | 1(14.3) | 44(51.2) |
| | KZ | 97 | 67(69.1) | 19 | 16(84.2) | 8 | 6(75) | - | - | 5 | 3(60) | - | - | 92(71.3) |
| | CRO | 101 | 59(58.4) | 17 | 12(70.6) | 8 | 3(38) | 5 | 5(100) | 4 | 1(25) | 7 | 7(100) | 87(58.4) |
| | CXM | 107 | 71(66.4) | 21 | 19(90.5) | 12 | 8(66.7) | - | - | 12 | 8(67) | - | - | 106(69.7) |
| | TZP | 46 | 9(20) | 7 | 5(71.4) | - | - | 10 | 8(80) | 8 | 4(50) | 4 | 0 | 26(34.7) |
| | CIP | 159 | 103(64.8) | 29 | 10(34.5) | 11 | 9(73) | 12 | 11(92) | 11 | 8(73) | 7 | 3(43) | 144(62.8) |
| | AUG | 113 | 69(61.1) | 24 | 16(66.7) | 12 | 10(83) | - | - | - | - | - | - | 95(63.8) |
| Carbapenem | IMP | 60 | 2(3.3) | 13 | 2(15.4) | 5 | 2(40) | 7 | 5(71.4) | 9 | 2(22.2) | 3 | 0 | 13(13.4) |
| Sulfonamides | SXT | 117 | 91(77.8) | 20 | 16(80) | 11 | 10(91) | 7 | 6(86) | 9 | 7(78) | - | - | 130(79) |
| Tetracycline | TE | 63 | 46(73) | 10 | 8(80) | 10 | 7(70) | 8 | 5(62.5) | 11 | 10(91) | - | - | 76(74.5) |
| Nitrofurans | FM | 77 | 11(14) | 13 | 5(38.5) | 12 | 1(8.3) | - | - | 7 | 3(43) | - | - | 20(18.3) |
| Over all AMR | | 1231 | 629(51.1) | 234 | 139(59.4) | 114 | 73(64) | 83 | 69(83) | 108 | 61(56) | 40 | 13(32.5) | |
| Overall AMR for the tested isolates | | Total tested isolates 1810 Resistance isolates 984 (54.4%) | | | | | | | | | | | | |

Key: Amikacin (AMK), Gentamicin (GEN), Piperacillin (PIP), Ceftazidime (CAZ), Cefazolin (KZ), Ceftriaxone (CRO), piperacillin + Tazobactam (TZP), Cefuroxime (CXM), Ciprofloxacin (CIP), Augmentin/ Amoxicillin-clavunate (AUG), Imipenem (IMP), Trimethoprim-Sulfamethoxazole (SXT), Tetracycline (TE) and Nitrofurantoin (FM), T = Total number of tested isolates, R = resistance isolates in %.

current study provide insights into the antimicrobial resistance patterns in Ethiopia, which has limited antimicrobial resistance monitoring data available.

In this study, out of a total of 1012 urine samples, 325 (32.1%) had significant growth and had no bacterial growth in 687 (67.9%). The prevalence here is lower than in some other studies: 67.2% in Nigeria [31], 60% reported in Lafia, Nigeria [32], 35.5% in Jos Nigeria [33]. The reason for the lower relative prevalence in this study could be that the patient was treated with antibiotics before arriving at the institute for diagnosis and the pathogens were killed or inhibited. The prevalence of this study was higher than other studies, 22% in Ibadan, Nigeria [34] and 22.7%, 28.5% in Ethiopia [35,36]. Study design, target population, sample size, and geographic variation may cause these differences.

The highest age-specific prevalence was found in the 21-30-year-old age group 73 (22.5%), which is consistent with other studies, showing that sexually active groups, especially women, are more likely to develop urinary tract infections [35,36]. This study is consistent with the studies recorded in India [2] and Ethiopia [36].

This study shows that Gram-negative bacteria (61%) are more common than Gram-positive bacteria (19.3%). Similar findings have been reported in Ethiopia and elsewhere [37–39]. According to this study, E. coli is the main urinary tract pathogen, which is consistent with almost all similar investigations [35,36,40]. This is due to the existence of many virulence factors dedicated to colonization and invasion of the urinary epithelium. Bacterial urinary tract pathogens isolated from UTI patients indicate extremely high levels of single and multiple antimicrobial resistance to commonly used drugs.

Sixty-five isolates (85.5%) of *Escherichia coli* are sensitive to nitrofurantoin and the rate of resistance to the drug is 14.5%. This result is consistent with other studies [41,42]. Other

**Table 5. Level of resistance of uropathogens bacterial isolates from urine specimen.**

| Isolates | Level of resistance n, (%) | | | | | | | | | | | | | |
|---|---|---|---|---|---|---|---|---|---|---|---|---|---|---|
| | R0 | R1 | R2 | R3 | R4 | R5 | R6 | R7 | R8 | R9 | R10 | MDR | XDR | PDR |
| *Escherichia coli(168)* | 23 | 14 | 32 | 24 | 19 | 23 | 18 | 8 | 4 | 3 | 0 | 99(59) | 30(18) | 4(2.4) |
| *Klebsiella pneumoniae(22)* | 1 | 0 | 4 | 3 | 3 | 2 | 6 | 2 | 1 | 0 | 0 | 17(77) | 9(41) | 0 |
| *Klebsiella oxytoca (8)* | 0 | 0 | 2 | 1 | 2 | 0 | 2 | 0 | 0 | 1 | 0 | 6(75) | 3(37.5) | 0 |
| *Klebsiella ozaniae (2)* | 1 | 1 | 0 | 0 | 0 | 0 | 0 | 0 | 0 | 0 | 0 | 0 | 0 | 0 |
| *Enterobacter spp(13)* | 2 | 1 | 2 | 0 | 4 | 2 | 2 | 1 | 1 | 0 | 0 | 10(77) | 4(31) | 1(7.7) |
| *Acinetobacter spp(12)* | 0 | 0 | 2 | 2 | 2 | 3 | 1 | 1 | 0 | 1 | 1 | 11(91.7) | 7(58.3) | 3(25) |
| *Citrobacter spp(12)* | 2 | 0 | 1 | 0 | 3 | 1 | 3 | 1 | 0 | 1 | 0 | 9(75) | 2(17) | 1(8.3) |
| *Pseudomonas spp(8)* | 3 | 4 | 1 | 1 | 0 | 0 | 0 | 0 | 0 | 0 | 0 | 1(12.5) | 0 | 0 |
| *Proteus mirablis(2)* | 0 | 1 | 0 | 0 | 0 | 1 | 0 | 0 | 0 | 0 | 0 | 1(50) | 0 | 0 |
| *Proteus vulgaris (2)* | 0 | 0 | 2 | 0 | 0 | 0 | 0 | 0 | 0 | 0 | 0 | 0 | 0 | 0 |
| *Morganella morgani (2)* | 0 | 0 | 0 | 1 | 0 | 0 | 0 | 1 | 0 | 0 | 0 | 1(50) | 1(50) | 1(50) |
| *Edwardsella spp(1)* | 0 | 0 | 0 | 0 | 1 | 0 | 0 | 0 | 0 | 0 | 0 | 1(100) | 0 | 0 |
| Total (256) | 32 | 21 | 46 | 32 | 34 | 32 | 32 | 14 | 6 | 6 | 1 | 156(61) | 56(22.4) | 10(4) |
| *CoNS(29)* | 5 | 4 | 3 | 7 | 6 | 3 | 0 | 0 | 0 | 1 | | 17(58.6) | 0 | 1(3.5) |
| *Enterococcus spp(16)* | 2 | 4 | 3 | 2 | 3 | 2 | 0 | 0 | 0 | 0 | 0 | 7(43.8) | 3(18.75) | 0 |
| *S.aureus (10)* | 1 | 0 | 1 | 1 | 3 | 0 | 1 | 3 | 0 | 0 | 0 | 8(80) | 6(60) | 0 |
| *S.saprophyticus (4)* | 0 | 1 | 1 | 0 | 0 | 1 | 1 | 0 | 0 | 0 | 0 | 2(50) | 1(25) | 0 |
| *S.agalactiae(4)* | 1 | 1 | 2 | 0 | 0 | 0 | 0 | 0 | 0 | 0 | 0 | 0 | 0 | 0 |
| Total (63) | 9 | 10 | 10 | 10 | 12 | 6 | 2 | 3 | 0 | 1 | 0 | 34(53) | 10(16) | 1(1.6) |

Note: R0: sensitive for all classes of antibiotics, R1: resistant for one class of antibiotics, R2: resistant for two classes of antibiotics, R3: resistant for three classes of antibiotics etc., MDR-multidrug resistant, XDR-Extreme Drug Resistant, PDR-Pan drug-resistant.

strains are sensitive to gentamicin and the drug resistance rate is 108 (69%). Compared with other studies and other studies from Bahir Dar, Ethiopia, this finding confers a higher sensitivity rate [37,39]. For trimethoprim/sulfamethoxazole resistance, our results also showed the resistance is much higher than previous studies [43].

*Pseudomonas aeruginosa* was more sensitive to piperacillin 3(100%), amikacin 2(100%), ciprofloxacin 4(57%) and ceftazidime 6(85%). This study has a similar finding with another study in Ethiopia [44,45].

In this study, the overall resistance rate of Gram-positive isolates was 51%, which is consistent with the studies documented in Ethiopia [44,46]. Among the isolated gram-positive bacteria, *Staphylococcus aureus* represents high drug resistance. This finding is consistent with studies conducted in Saudi Arabia [36], Ethiopia [35] and Iraq [31].

In this study, the estimated MDR isolates of GNB were 156(61%) which is higher than the study documented in Tigray [38], in Addis Ababa cancer patients [47]. In this study, 22.4% and 4% of the gram-negative isolates were XDR and PDR respectively, which is lower than the study conducted in Dessie [37], and in the Tigray region [38]. The reason for this observed high resistance may be due to the increasing irrational use of antibiotics, the transmission of resistance genes between people and people or /and animals to people, and consumption of animal products that treated antibiotics. Self-medication and non-compliance with medication and sales of the substandard drug may account for the rise in antibiotic resistance observed in this study. The clinical and financial burden to patients and healthcare providers for MDROs is challenging. Patients who are infected with MDROs often have an increased risk of prolonged illness and mortality. The cost of care for these patients can be more than double as

compared to those without MDRO infection. Regarding public health attention, MDROs are described as superbugs having very limited treatment options.

These findings have important therapeutic implications for the treatment and management of urinary tract infections (UTIs), especially those caused by multidrug-resistant urinary tract pathogens. Clinicians should first be aware that UTI patients are more likely to be infected with common urinary tract pathogens or relatively rare isolated urinary tract pathogens. Second, the high rate of multidrug resistance observed in this study is a major problem in the treatment of urinary tract infections (UTIs), and a systematic approach is needed to reduce the rate of resistance to antibiotics or minimize the use of broad-spectrum antibiotics [48].

Finally, in the case of multidrug resistance, rapid diagnostic tests (point-of-care tests) for timely, targeted treatment are the top priority. There is also a need for a drug control system that optimizes drug use and allows for a personalized approach to proposed treatments [48].

We believe that urinary tract infection (UTI) is an accessible goal for the development of health education programs aimed at reducing the prevalence of diseases in the community and improving the quality of life of patients in low- and middle-income countries.

## Limitation of the study

Our findings may not be inferred to specialized groups like HIV positive individuals or pregnant women and it was not possible to include anaerobic bacteria isolates due to lack of cultures media. Because of the lack of molecular methods and primes, antibiotic resistance encoding genes (ARGs) of isolates were not detected as a confirmatory test.

## Conclusion

Levels of urinary tract infections are high, and Gram-negative bacteria are the most common cause of urinary tract infections. It is found that the resistance to commonly used antibiotics is very high. This shows that drug resistance is a deep-rooted problem in Ethiopia. Inadequate treatment regimens, insufficient patient adherence, and uncontrolled distribution and trade of drugs, as well as lack and poor quality of antibiotics, can also cause antibiotic resistance.

To this end, the incidence of urinary tract infections should be minimized and the susceptibility of specific pathogens to commonly used antibacterial agents should be continuously monitored. Therefore, based on this, there is a need to provide improved, adequate and affordable health services in the community, especially reproductive health services. Finally, a nationwide survey and study on antibiotic resistance are needed to assess this devastating nationwide situation and formulate control strategies.

## Supporting information

**S1 Dataset. Dataset used for analysis of the result.**
(SAV)

**S2 Dataset. Dataset used for the analysis of MDR, XDR and PDR.**
(XLSX)

## Acknowledgments

The authors would like to acknowledge Mr. Kassahun Habtamu for his professional support.

Special thanks to all staff of the Ethiopian Public Health Institution Clinical Bacteriology and Mycology department for their valuable support.

## Author Contributions

**Conceptualization:** Tesfa Addis, Yonas Mekonnen, Zeleke Ayenew, Surafel Fentaw, Habtamu Biazin.

**Data curation:** Tesfa Addis, Yonas Mekonnen, Zeleke Ayenew, Surafel Fentaw, Habtamu Biazin.

**Formal analysis:** Tesfa Addis, Yonas Mekonnen, Zeleke Ayenew, Habtamu Biazin.

**Funding acquisition:** Tesfa Addis, Yonas Mekonnen, Surafel Fentaw.

**Investigation:** Tesfa Addis, Yonas Mekonnen, Zeleke Ayenew, Surafel Fentaw.

**Methodology:** Tesfa Addis, Yonas Mekonnen, Surafel Fentaw, Habtamu Biazin.

**Project administration:** Tesfa Addis, Yonas Mekonnen.

**Resources:** Tesfa Addis, Yonas Mekonnen.

**Software:** Tesfa Addis, Yonas Mekonnen, Zeleke Ayenew, Surafel Fentaw, Habtamu Biazin.

**Supervision:** Tesfa Addis, Yonas Mekonnen, Zeleke Ayenew, Surafel Fentaw.

**Validation:** Yonas Mekonnen, Zeleke Ayenew, Habtamu Biazin.

**Visualization:** Surafel Fentaw, Habtamu Biazin.

**Writing – original draft:** Zeleke Ayenew, Habtamu Biazin.

**Writing – review & editing:** Zeleke Ayenew, Habtamu Biazin.

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
