## [Decision Letter · Decision Letter 0]

4 Oct 2021

PONE-D-21-27426The Burden of Antimicrobial resistance pattern in uropathogens infection referred to Ethiopian Public Health InstitutePLOS ONE

Dear Dr. Kebede,

Thank you for submitting your manuscript to PLOS ONE. After careful consideration, we feel that it has merit but does not fully meet PLOS ONE’s publication criteria as it currently stands. Therefore, we invite you to submit a revised version of the manuscript that addresses the points raised during the review process.

ACADEMIC EDITOR: A major revision is needed; please address all the comments and suggestions that were raised by the reviewers. 

We look forward to receiving your revised manuscript.

Kind regards,

Abdelazeem Mohamed Algammal, Prof, Ph.D

Academic Editor

PLOS ONE

Journal Requirements:

Reviewers' comments:

Reviewer's Responses to Questions

**Comments to the Author**

1. Is the manuscript technically sound, and do the data support the conclusions?

Reviewer #1: Partly

Reviewer #2: Partly

2. Has the statistical analysis been performed appropriately and rigorously? 

Reviewer #1: No

Reviewer #2: No

3. Have the authors made all data underlying the findings in their manuscript fully available?

Reviewer #1: Yes

Reviewer #2: No

4. Is the manuscript presented in an intelligible fashion and written in standard English?

Reviewer #1: No

Reviewer #2: Yes

5. Review Comments to the Author

Reviewer #1: Comments to authors:

- The current study is interesting; however, the authors should address the following comments to improve the quality of the manuscript:

- The manuscript should be revised for language editing and grammar mistakes.

Title:

I think the work would benefit from the title that contains the main conclusion of the study (should be derived from the conclusion). Please modify the title.

Abstract:

- The abstract must illustrate the used methods and the most prevalent results (give more hints about methods and results). Besides, rephrase the main conclusion of your findings.

Introduction:

-Give a hint about the virulence factors of E. coli, K. pneumonia, and Staphylococcus aureus and the mechanism of disease occurrence.

- The authors should illustrate the public health importance concerning the emergence of multidrug-resistant (MDR) bacterial pathogens that reflecting the necessity of new potent and safe antimicrobial agents. Several studies proved the widespread MDR- bacterial pathogens;

Authors could add the following paragraph:

Multidrug resistance has been increased all over the world that is considered a public health threat. Several recent investigations reported the emergence of multidrug-resistant bacterial pathogens from different origins including humans, poultry, cattle, and fish that increase the need for routine application of the antimicrobial susceptibility testing to detect the antibiotic of choice as well as the screening of the emerging MDR strains. You should cite the following valuable studies:

1.PMID: 33177849

2.PMID: 32497922

3.PMID:33061472

4.PMID: 33947875

5.PMID: 32472209

6.PMID: 32994450

7.PMID: 33188216

8.PMID: 32235800

9. Abouelmaatti, R. et al. (2013): Cloning and analysis of Nile tilapia Toll-like receptors type-3 mRNA: Centr. Eur. J. Immunol; 38 (3): 277-282.

DOI: https://doi.org/10.5114/ceji.2013.3774020

-Rephrase the aim of the work to be clear and better sound.

Material and methods

-Add the following title to the Methods section:

-Isolation and identification of Bacterial pathogens incriminated in urinary tract infection:

• Discuss in details the methods of isolation and identification of various bacterial pathogens. Besides, specific reference should be added.

•Add the company, city, and country of the used bacterial media and reagents that were used in the biochemical identification of isolates. Also, enumerate all used biochemical reactions.

- Antimicrobial susceptibility testing:

•Illustrate the antimicrobial classes of the tested antimicrobial agents.

•A specific reference should be added; such as CLSI 2018.

•The authors are advised to classify the tested isolates to MDR or XDR as described by Magiorakos et al.

Magiorakos AP, Srinivasan A, Carey RB, Carmeli Y, Falagas ME, Giske CG, et al. Multidrug-resistant, extensively drug-resistant and pandrug-resistant bacteria: An international expert proposal for interim standard definitions for acquired resistance. Clin Microbiol Infect. 2012; 18:268–81. doi:10.1111/j.1469-0691.2011.03570.x.

-The authors are advised to perform PCR-based detection of the antimicrobial resistance genes in multidrug resistant strains if applicable. (Or address this point to the study limitations)

- illustrate the details of the used software in the the statistical analyses?

-Result: (Correct the title to be: Results)

-Illustrate the phenotypic characteristics of different recovered bacterial pathogens.

-Illustrate in a new table the occurrence of MDR (Multidrug resistance) among the recovered isolates (illustrate the names of the antimicrobial classes and different antibiotics):

No. of strains%Type of resistance

R OR MDR OR XDRPhenotypic multidrug resistance

(Antimicrobial classes and different antibiotics).

The antibiotic -resistance genes

-The authors are advised to perform PCR-based detection of the antimicrobial resistance genes in the multidrug resistant strains if applicable. (Or address this point to the study limitations).

-Where are the results of the statistical analyses? (Illustrate the P- value in the tables and the significance of your findings).

-Discussion:

- The authors are advised to illustrate the real impact of their findings without repetition of results.

-Conclusion

- Should be rephrased to be sounded. A real conclusion should focus on the question or claim you articulated in your study, which resolution has been the main objective of your paper?

Reviewer #2: - The current study has a significant impact, but it needs a major revision:

- The manuscript should be revised for grammar mistakes.

- Please write the scientific names of all pathogens in italic form all over the manuscript.

-The title is broad, please modify the title.

- Add more details about the used methods and most prevalent results in the abstract.

-In the introduction: discuss the public health importance of the recovered bacterial pathogens and different infection caused by them.

-Improve the aim of work.

Methods:

-Where are the methods of isolation and identification?? Specific references should be added to all the used methods and techniques.

-Add the manufacturing company, city, and country for the used media and antimicrobial discs. Add specific references.

-Why you ignored the detection of antibiotic resistance genes??

-Evaluate the significance of your findings using the statistical analysis.

-Results:

- Discuss in detail the phenotypic characters of the isolated bacterial pathogens.

-Why you ignored the detection of antibiotic resistance genes??

-Where are the results of the statistical analyses?

-Discussion:

- Please improve.

-Please improve the main conclusion of the manuscript.

6. PLOS authors have the option to publish the peer review history of their article (what does this mean?). If published, this will include your full peer review and any attached files.

Reviewer #1: No

Reviewer #2: No

---

## [Author Response · Author response to Decision Letter 0]

20 Oct 2021

Date: 05/10/2021

Response to Reviewers

To reviewers and journal editors:

Dear reviewers,

We appreciate for taking your precious time and forwarding your valuable comments/concerns, which have significantly improved our application. We are also grateful for this positive feedback. Please find below our response to each point raised (in bold), showing how the manuscript has been amended in response to each comment.

We are looking forward to hearing from you in due course,

Sincerely, 

Habtamu Biazin (Corresponding author)

Title of the manuscript: The Burden of Antimicrobial resistance pattern in uropathogens infection referred to Ethiopian Public Health Institute  

Manuscript Number: PONE-D-21-27426

Below we provide the point-by-point responses. All modifications in the manuscript have been highlighted in red.

Reviewer #1: Comments to authors: 

Comments: The current study is interesting; however, the authors should address the following comments to improve the quality of the manuscript:

Response: Thank you very much for this positive feedback.

Comments: The manuscript should be revised for language editing and grammar mistakes.

Response: We are grateful for the suggestion. We went through the entire manuscript to eliminate grammatical and editing mistakes.

Comments: I think the work would benefit from the title that contains the main conclusion of the study (should be derived from the conclusion). Please modify the title.

Response: Thank you for pointing this out. We agree with this comment. Therefore, we have amended the title as follow: “Bacterial uropathogens and burden of Antimicrobial resistance pattern in urine specimens referred to Ethiopian Public Health Institute.”

Abstract: 

Comments: The abstract must illustrate the used methods and the most prevalent results (give more hints about methods and results). Besides, rephrase the main conclusion of your findings.

Responses: We agree with this and have incorporated your comments (page 1 line17&18, 20-22; 36, 37, 40-42).

Introduction: 

Comments: Give a hint about the virulence factors of E. coli, K. pneumonia, and Staphylococcus aureus and the mechanism of disease occurrence.

Responses: Thank you for this suggestion. It would have been interesting to explore this aspect. However, in the case of our study, it seems slightly out of scope because this study mainly explores the magnitude of bacterial pathogens incriminated in UTIs and their antibiotic susceptibility of the isolates. It may be interesting if the study focuses on the virulence factors and associated factors of the pathogenesis of UTI. Still, we tried to add the following suggestion on page 3 (Line 68-73). 

Comments: The authors should illustrate the public health importance concerning the emergence of multidrug-resistant (MDR) bacterial pathogens that reflecting the necessity of new potent and safe antimicrobial agents. Several studies proved the widespread MDR- bacterial pathogens;

Responses: Agree. We have made the following changes on pages three & four (paragraph7, lines 90-102).

Comments: Authors could add the following paragraph: Multidrug resistance has been increased all over the world that is considered a public health threat. Several recent investigations reported the emergence of multidrug-resistant bacterial pathogens from different origins including humans, poultry, cattle, and fish that increase the need for routine application of the antimicrobial susceptibility testing to detect the antibiotic of choice as well as the screening of the emerging MDR strains. You should cite the following valuable studies: 

1. PMID: 33177849

2. PMID: 32497922

3. PMID: 33061472

4. PMID: 33947875

5. PMID: 32472209

6. PMID: 32994450

7. PMID: 33188216

8. PMID: 32235800

9. Abouelmaatti, R. et al. (2013): Cloning and analysis of Nile tilapia Toll-like receptors type-3 mRNA: Centr. Eur. J. Immunol; 38 (3): 277-282. DOI:https://doi.org/10.5114/ceji.2013.3774020

Responses: We agree with this and have incorporated your suggestion in the revised manuscript on page 3 (line 90-102).

Comments: Rephrase the aim of the work to be clear and better sound.

Responses: Thank you very much for pointing this out. We revised the objective as follows: “Therefore, this study aimed to determine the magnitude of bacterial isolates, and antibiotic resistance pattern of the urinary pathogens isolated from patients referred to EPHI.” Page 4 of the revised manuscript

Material and methods

Comments: Add the following title to the Methods section: Isolation and identification of Bacterial pathogens incriminated in urinary tract infection: Discuss in detail the methods of isolation and identification of various bacterial pathogens. Besides, specific references should be added.

Responses: We agree with the reviewer’s assessment. Accordingly, we have revised this section and added the suggested comments as follow (Page 5 and 6; Page 7 paragraphs, line147-169. 

The text read as:

“Isolation, Identification and Characterization of bacteria causing UTIs

Isolation of pathogens: The pathogens have been isolated from urine sediments. The urine sample was shaken well to resuspend the organisms and 10 ml was decanted into a centrifuge tube. The tube was kept closed to avoid contamination(25). The sample was centrifuged at 2000 rpm for 10 minutes. The entire sample was decanted, but 0.5 ml of sediment from the tube was suspended with a sterile metal loop. A loop of the sediment was inoculated into a tube with a medium(26).

Identification of pathogens: 

Cultural observation: Color, size, and colony morphology are observed from the incubated plates.

Microscopic Examination of urine specimen: Slides were prepared from each different colony observed on the plates and Gram staining was performed. Results such as arrangement, gram reaction (gram-positive or gram-negative), and shape of bacteria are seen from the investigations(25,27)

Microbiological analysis of urine specimen:

Uropathogens were identified by inoculating /streaking of urine samples on various selective and differential media such as CLED agar, blood agar, chocolate agar, MacConkey agar based on their colour morphology after an incubation time of 18-24 hours at 37℃ (26).

Biochemical Examination: 

The selected colonies, based on culture, microscopic and microbiological examinations, were subjected to biochemical examinations(Oxoid Ltd, Basingstoke, UK) (carbohydrate utilization test, triple sugar iron agar test or Kligler iron agar), oxidase test, catalase test, nitrate reduction test, indole production test, methyl red test, Voges-Proskauer test, citrate recovery test, urease test) for the detection of the pathogen(27,28)

Comments: Add the company, city, and country of the used bacterial media and reagents that were used in the biochemical identification of isolates. Also, enumerate all used biochemical reactions.

Responses: We inserted the suggested comments accordingly.

Comments: Antimicrobial susceptibility testing: Illustrate the antimicrobial classes of the tested antimicrobial agents.

Responses: Thank you for this suggestion. The antimicrobial classes of the tested antimicrobial agents are indicated in Tables 3 and 4 for each type of isolated. 

Comments: A specific reference should be added; such as CLSI 2018.

Responses: Thank you. It is accepted and revised accordingly.

Comments: The authors are advised to classify the tested isolates to MDR or XDR as described by Magiorakos et al. Magiorakos AP, Srinivasan A, Carey RB, Carmeli Y, Falagas ME, Giske CG, et al. Multidrug-resistant, extensively drug-resistant and pan drug-resistant bacteria: An international expert proposal for interim standard definitions for acquired resistance. Clin Microbiol Infect. 2012; 18:268–81. doi:10.1111/j.1469-0691.2011.03570.x.

Responses: Thank you for this suggestion. It would have been interesting to explore this aspect and we incorporated the comment in table 5 page 11.

 Comments: The authors are advised to perform PCR-based detection of the antimicrobial resistance genes in multidrug-resistant strains if applicable. (On the other hand, address this point to the study limitations)

Responses: We agree that this is a potential limitation of the study. We have added this as a limitation on page 14 of the revised manuscript. The revised sentence is as follows (Line 345 and 346). “Because of the lack of molecular methods and primes, antibiotic resistance encoding genes (ARGs) of isolates were not detected as a confirmatory test”.

Comments: illustrate the details of the used software in the statistical analyses?

Responses: We think this is an excellent suggestion. However, we believe that these soft wares used for statistical analyses (EpiData v4.6 and SPSS v25) are open access soft wares everybody can access them. The instructions for use are freely available. Therefore, describing these statistical tools beyond the level of this detail that is presented on the test description is slightly excessive.

Comments: Result: (Correct the title to be Results)

Responses: accepted and amended accordingly. 

Comments: Illustrate the phenotypic characteristics of different recovered bacterial pathogens.

Responses: Thank you. The general phenotypic characteristics of the recovered isolates are described in the method sections on page 6, line 150.

Comments: Illustrate in a new table the occurrence of MDR (Multidrug Resistance) among the recovered isolates (illustrate the names of the antimicrobial classes and different antibiotics):

No. of strains percentage Type of resistance. R OR MDR OR XDR Phenotypic multidrug resistance (Antimicrobial classes and different antibiotics). 

Responses: Thank you for this suggestion. The antimicrobial classes of the tested antimicrobial agents are indicated in Tables 3 and 4 for each type of isolated (gram-positive and gram-negative bacterial isolates) while the MDR, XDR, and PDR are presented in table 5 page 14.

Comments: The antibiotic-resistance genes

Responses: We agree that this is a potential limitation of the study. We have added this as a limitation and stated as: “Because of the lack of molecular methods and primes, antibiotic resistance encoding genes (ARGs) of isolates were not detected as a confirmatory test.”

Comments: The authors are advised to perform PCR-based detection of the antimicrobial resistance genes in the multidrug-resistant strains if applicable. (Alternatively, address this point to the study limitations).

Responses: As we stated earlier this is a potential limitation of the study and we could not conduct it because of the lack of molecular methods and primes.

Comments: Where are the results of the statistical analyses? (Illustrate the P-value in the tables and the significance of your findings).

Responses: Thank you for pointing this out. Although we agree that this is an important

consideration, it is not appropriate for inclusion in this manuscript because this study did not aim to determine associated risk factors for the findings. Nevertheless, we tried to associate the age group and sex of the participants, which is not statistically associated. The result is tabulated in table 1 page 7.

Discussion: 

Comments: The authors are advised to illustrate the real impact of their findings without repetition of results.

Responses: Thanks for your kind reminders. We revised the discussion as follows (page 14, line 318-28).

Conclusion

 Comments: This should be rephrased to be sounded. A real conclusion should focus on the question or claim you articulated in your study, which resolution has been the main objective of your paper?

Responses: Revised according to the amended title of the manuscript.

Reviewer #2: The current study has a significant impact, but it needs a major revision: 

Responses: Thank you for this positive feedback.

Comments: The manuscript should be revised for grammar mistakes.

Responses: We are grateful for the suggestion. We went through the entire manuscript to eliminate grammatical and editing mistakes.

Comments: Please write the scientific names of all pathogens in italic form all over the manuscript.

Responses: We are grateful for the suggestion. We went through the entire manuscript to fix the suggested comments.

Comments: The title is broad, please modify the title

Responses: Thank you for pointing this out. We agree with this comment. Therefore, we have amended the title as follow: “Bacterial uropathogens and burden of Antimicrobial resistance pattern in urine specimens referred to Ethiopian Public Health Institute.”

Comments: Add more details about the used methods and most prevalent results in the abstract.

Responses: Thank you very much for the reminder. We have made revisions accordingly (line 17-22, page 1).

Comments: In the introduction: discuss the public health importance of the recovered bacterial pathogens and different infections caused by them.

Responses: Thanks for your kind reminders. We revised the introduction with the following sentences (lines 68-73, 90-102 and 115-118).

Comments: Improve the aim of the work.

Responses: Thank you very much for the reminder. We have made revisions accordingly.

Methods: 

Comments: Where are the methods of isolation and identification?? Specific references should be added to all the used methods and techniques.

Responses: We agree with the reviewer’s assessment. Accordingly, we have revised this section and added the suggested comments as follow (Page 5 and 6; Page 7 paragraphs, line147-169).

Comments: Add the manufacturing company, city, and country for the used media and antimicrobial discs. Add specific references.

Responses: Thank you very much for the reminder. We have made revisions accordingly.

Comments: Why you ignored the detection of antibiotic resistance genes?

Responses: We agree that this is a potential limitation of the study. We have added this as a limitation and stated as: “Because of the lack of molecular methods and primes, antibiotic resistance encoding genes (ARGs) of isolates were not detected as a confirmatory test.”

Comments: Evaluate the significance of your findings using the statistical analysis.

Responses: Thank you very much. We revised and included it in table 1.

-Results

-Comments: Discuss in detail the phenotypic characters of the isolated bacterial pathogens.

Responses: We revised and included them in the method section.

Comments: Why you ignored the detection of antibiotic resistance genes?

Responses: As we stated earlier this is a potential limitation of the study and we could not conduct it because of the lack of molecular methods and primes.

Comments: Where are the results of the statistical analyses?

Responses: Thank you very much. We revised and included it in table 1.

Discussion: 

Comments: Please improve.

Responses: Thanks for your kind reminders. We revised the discussion section (page 14, lines 318-28).

Comments: Please improve the main conclusion of the manuscript

Responses: Revised according to the amended title of the manuscript.

We look forward to hearing from you in due time regarding our submission and to respond to any further questions and comments you may have.

Sincerely,

Habtamu Biazin

---

## [Decision Letter · Decision Letter 1]

22 Oct 2021

Bacterial uropathogens and burden of Antimicrobial resistance pattern in urine specimens referred to Ethiopian Public Health Institute

PONE-D-21-27426R1

Dear Dr. Kebede,

We’re pleased to inform you that your manuscript has been judged scientifically suitable for publication and will be formally accepted for publication once it meets all outstanding technical requirements.

Kind regards,

Abdelazeem Mohamed Algammal, Prof, Ph.D

Academic Editor

PLOS ONE

Additional Editor Comments (optional):

Reviewers' comments:

Reviewer's Responses to Questions

**Comments to the Author**

1. If the authors have adequately addressed your comments raised in a previous round of review and you feel that this manuscript is now acceptable for publication, you may indicate that here to bypass the “Comments to the Author” section, enter your conflict of interest statement in the “Confidential to Editor” section, and submit your "Accept" recommendation.

Reviewer #1: All comments have been addressed

2. Is the manuscript technically sound, and do the data support the conclusions?

Reviewer #1: Yes

3. Has the statistical analysis been performed appropriately and rigorously? 

Reviewer #1: Yes

4. Have the authors made all data underlying the findings in their manuscript fully available?

Reviewer #1: Yes

5. Is the manuscript presented in an intelligible fashion and written in standard English?

Reviewer #1: Yes

6. Review Comments to the Author

Reviewer #1: The authors have carried out a significant changes to the manuscript. They have addressed all the suggested corrections and comments. Really, it's an interesting study that has a significant impact. Now, the manuscript could be accepted.

7. PLOS authors have the option to publish the peer review history of their article (what does this mean?). If published, this will include your full peer review and any attached files.

Reviewer #1: No

---

## [Editor Report · Acceptance letter]

29 Oct 2021

PONE-D-21-27426R1 

Bacterial uropathogens and burden of Antimicrobial resistance pattern in urine specimens referred to Ethiopian Public Health Institute 

Dear Dr. Kebede:

I'm pleased to inform you that your manuscript has been deemed suitable for publication in PLOS ONE. Congratulations! Your manuscript is now with our production department. 

Kind regards, 

on behalf of

Professor Abdelazeem Mohamed Algammal 

Academic Editor

PLOS ONE